

# Effects of variability in probable maximum precipitation patterns on flood losses

Andreas Paul Zischg[1,2], Guido Felder[1], Rolf Weingartner[1], Niall Quinn[3], Gemma Coxon[2], Jeffrey Neal[2], Jim Freer[2], Paul Bates[2]

[1]University of Bern, Institute of Geography, Oeschger Centre for Climate Change Research, Mobiliar Lab for Natural Risks, Bern, 3012, Switzerland
[2]School of Geographical Sciences, University of Bristol, Bristol, BS8 1SS, United Kingdom
[3] Fathom Ltd., Bristol, BS1 6QF, United Kingdom

*Correspondence to*: Andreas Paul Zischg (andreas.zischg@giub.unibe.ch)

**Abstract.** The assessment of the impacts of extreme floods is important for dealing with residual risk, particularly for critical infrastructure management and for insurance purposes. Thus, modelling of the probable maximum flood (PMF) from probable maximum precipitation (PMP) by coupling hydrologic and hydraulic models has gained interest in recent years. Herein, we examine whether variability in precipitation patterns exceeds or is below other uncertainties in flood loss estimation and if the flood losses within a river basin are related to the probable maximum discharge at the basin outlet. We developed a model experiment with an ensemble of probable maximum precipitation scenarios created by Monte-Carlo simulations. For each rainfall pattern, we computed the flood losses with a model chain and benchmarked the effects of variability in rainfall distribution with other model uncertainties. The results show that flood losses vary considerably within the river basin and depend on the timing and superimposition of the flood peaks from the basin's sub-catchments. In addition to the flood hazard component, the other components of flood risk, exposure and vulnerability, contribute remarkably to the overall variability. This leads to the conclusion that the estimation of the probable maximum expectable flood losses in a river basin should not be based exclusively on the PMF. Consequently, the basin-specific sensitivities to different precipitation patterns and the spatial organisation of the settlements within the river basin need to be considered in the analyses of probable maximum flood losses.

## 1 Introduction

Floods are one of the most damaging natural hazards, accounting for a majority of all economic losses from natural events worldwide (UNISDR, 2015). Managing flood risks requires knowledge about hazardous processes and the impacts of floods. Typically the impacts of design floods with a certain (extreme) return period (IPCC, 2012), or the impacts of worst-case floods are required for sound risk analysis and for the planning of risk reduction measures. In particular, for portfolio risk analyses of insurance companies, the estimation of the probable maximum loss is important for fulfilling financial regulations and stability criteria. Furthermore, critical infrastructure, such as power stations, has to be protected against





extreme floods. Since floods are expected to increase due to climatic changes (Asadieh and Krakauer, 2015; Arnell and Gosling, 2016; Beniston et al., 2007; Bouwer, 2013; Fischer and Knutti, 2016; Millán, 2014; Pfahl et al., 2017; Rajczak et al., 2013; Scherrer et al. 2016), flood risk analyses and the management of extreme events will become even more relevant (Smolka, 2006; Yuan et al., 2017). Hence, insurance companies as well as governmental institutions are increasingly

interested in quantifying flood risks, and especially in estimating the impacts of probable maximum floods leading to high cumulative losses (Burke et al., 2016; Morrill and Becker, 2017) or the destruction of critical infrastructure (Hasan and Foliente, 2015; Mechler et al., 2010; Michaelides, 2014).

An important aspect in flood risk analysis is the modelling of worst-case floods and their impacts (Büchele et al., 2006). One main question herein is the search for the upper physical limits of discharge in a river basin, i.e. the maximum outflow from

a catchment that is possible with the given catchment characteristics and the maximum rainfall in the climate region (Felder and Weingartner, 2017). Here, the hydrologic modelling to derive probable maximum floods (PMF) from probable maximum precipitation (PMP) is an important first step as a basis for inundation modelling (Felder et al., 2017). The PMP is defined as "the theoretical maximum precipitation for a given duration under modern meteorological conditions" (World Meteorological Organization, 2009). Instead, the probable maximum flood (PMF) is defined as "the theoretical maximum

flood that poses extremely serious threats to the flood control of a given project in a design watershed" (World Meteorological Organization, 2009). The PMF is estimated on the basis of the PMP and is commonly used in practice for the planning of hydropower dams. However, there is still a controversial discussion on the underlying concept of PMP, particularly on the assumption that the upper tail of flood distributions is bounded (Micovic et al., 2015). Comprehensive summaries of this discussion are provided by (Salas et al., 2015) and by (Rouhani and Leconte, 2016). Nevertheless,

PMP/PMF estimation methods are continuously evolved and improved. Beauchamp et al. (2013), Lagos-Zuniga and Vargas M., (2014), and Felder and Weingartner (2016) discuss the role of the spatio-temporal distribution of the PMP on the PMF, while Rousseau et al. (2014) and Stratz and Hossain (2014) discuss climate change and stationarity issues. Hence, Faulkner and Benn (2016), Micovic et al. (2015), Rouhani and Leconte (2016), and Salas et al. (2014) propose to incorporate uncertainty bands into the PMP estimation.

Nevertheless, the detailed triggering mechanism and the temporal evolution of large flood events, specifically of worst-case scenarios, are not yet fully understood. An important question concerns how the peak discharge and the volume of a flood depend on the intensity and track of the triggering precipitation events, i.e. the spatio-temporal pattern of precipitation (Adams et al., 2012; Bruni et al., 2015; Cristiano et al., 2017; Emmanuel et al., 2015, Emmanuel et al., 2016; Ochoa-Rodriguez et al., 2015; Paschalis et al., 2014; Rafieeinasab et al., 2015; Zhang and Han, 2017). In addition to the storm track

dynamics, the peak flow depends on the watershed characteristics (Singh, 1997). In mountainous catchments with a high topographical complexity, the storm track and the precipitation pattern are influenced by the mountain ranges. Furthermore, the river network is influenced by geological and tectonic structures and is thus more complex in mountainous terrain than in low-lying areas. Thus, in upland areas a high variability in the spatio-temporal pattern of a probable maximum precipitation event and the resulting river flows has to be assumed. The definition of the spatio-temporal characteristics of PMP scenarios





is a crucial step in the analysis of the impacts of extreme flood events. Hence, different approaches in distributing PMP in space and time over a catchment have been developed recently (Beauchamp et al., 2013; Dodov and Foufoula-Georgiou, 2005; Foufoula-Georgiou, 1989; Franchini et al., 1996, Felder and Weingartner, 2016). Regarding mountainous meso-scale catchments with an area of a few thousand km$^2$, insights into precipitation patterns leading to the most extreme floods are

rather rare. The precipitation pattern leads to a specific pattern of the outflows from the sub-catchments. Depending on the geometry of the main river network, this timing of the outflows from the sub-catchments influences peak discharge in the individual river reaches. Hence, the relative timing of peak discharge arrivals in river confluences as a consequence of the spatio-temporal distribution of the rainfall pattern has to be addressed (Nicótina et al., 2008; Nikolopoulos et al., 2014; Pattison et al., 2014; Emmanuel et al., 2016; Zoccatelli et al., 2011). Thus, a sound analysis of extreme floods in a complex

river basin requires an assessment of the variability of chronological superimpositions of flood waves in tributaries and the effect of this on the probability of inundation. Neal et al. (2013) highlight the importance of spatial dependence between tributaries in terms of inundation probability and magnitude. Consequently, the amount of flood losses are also expected to vary with the timing of peak flows in the tributaries. Ochoa-Rodriguez et al. (2015) also stated that the temporal variation of rainfall inputs affects hydrodynamic modelling results remarkably. Emmanuel et al. (2015) showed that the spatio-temporal

organization of rainfall plays an important role in the discharge at the outlet of the catchment and stated that a simulation approach is needed to study the effects of rainfall variability in complex river basins. The effects vary with the catchment size and its characteristics. Nevertheless, they state that there is a knowledge gap in this field. Probably the study that is most clearly focused on the role of the tributary relative timing and sequencing for extreme floods is presented by Pattison et al. (2014). They showed that tributary relative timing and synchronization is important in the determination of flood peak

downstream. Thus, the distribution of extreme rainfall in space and time must play a critical role in determining the probable maximum flood (PMF) and the peak discharge at the catchment outlet.

While the influence of rainfall variability on catchment response is under investigation, the further influence on flood losses is rarely investigated. To our knowledge, so far only Sampson et al. (2014) analysed in-depth the effects of different precipitation scenarios on flood losses. However, the Sampson *et al.* study focused on an urban area and on a (relatively)

small scale. Hitherto, no studies were conducted in mountainous river basins to our knowledge.

In addition to the variability in precipitation patterns, other uncertainties have to be considered in flood loss estimation. Besides uncertainties in hydrological modelling that are not considered in this study, other factors lead to uncertainties in inundation modelling and in flood loss estimation. Uncertainties in inundation modelling and flood risk analysis are addressed by Apel et al. (2008), Di Baldassarre et al. (2010), Gai et al. (2017), Merz and Thieken (2009), and Neal et al.

(2013). Savage et al. (2015) and Fewtrell et al. (2008) describe the effects of spatial scale on inundation modelling. Altarejos-García et al. (2012), Chatterjee et al. (2008), Horritt and Bates (2001), Horritt and Bates (2002), Kvočka et al. (2015), and Neal et al. (2012b) discuss the effects of the chosen inundation model, its parametrization, and the role of input data on flood modelling results. Other uncertainties in flood modelling outputs are related to uncertainties in levee heights (Sanyal, 2017), or in the digital elevation models (Saksena and Merwade, 2015). Beside the uncertainties in flood modelling,





observational uncertainties also need to be recognized with recent studies highlighting the importance of observational errors in rainfall and discharge data (McMillan et al., 2012; Coxon et al., 2015).

Furthermore, uncertainties in the economic models used to estimate flood losses and flood damages are relevant (Moel et al., 2015). Herein, the input data, the choice of the impact indicators, the scale and the vulnerability models are relevant sources

of uncertainty (Ward et al., 2013; Apel et al., 2008; Ward et al., 2013; Merz and Thieken, 2009; Moel and Aerts, 2011). In particular, vulnerability functions are considered as one of the most relevant sources of uncertainty in flood loss estimation (Ward et al., 2013; Sampson et al., 2014). Thus, uncertainty analysis is a key aspect in flood risk assessment. Some of the limitations and uncertainties mentioned above are addressed by several recent studies. Especially in a framework of coupled models, sensitivity analysis is important to assess the propagation of cascading uncertainties to the final result (Ward et al.,

2013; Rodríguez-Rincón et al., 2015) However, sensitivity analyses and uncertainty analyses of coupled models or model chains are rarely investigated topics.

In summary, we identify a research gap in our understanding of the effects of spatio-temporal precipitation patterns on the amount of flood losses in a river basin. The main goals of this study are to analyse the effects of variability in probable maximum precipitation patterns on flood losses, and to compare these effects with other uncertainties in flood loss modelling

in a complex mountain catchment (i.e. choice of inundation models or vulnerability functions). One important question is whether the variability in precipitation patterns is more or less influential than other uncertainties in flood loss estimation. A second question is whether the maximum discharge at the catchment outlet is a reliable proxy indicator for identifying the scenario(s) for worst case flood loss.

## 2 Methods

To address the above questions using a numerical experiment we constructed an inundation modelling framework composed of several coupled modules. The model chain was developed for the Aare River basin in Switzerland (3000 km$^2$) and consists of five main components: a precipitation module, a hydrology module, a hydrodynamic routing module, a hydrodynamic inundation module and a damage module. In the following, the setup of the model chain is described. The model chain was used in the framework of a model experiment to assess the uncertainties in the model input, i.e. the spatio-

temporal variability in precipitation patterns. These uncertainties were subsequently compared with other uncertainties, i.e. uncertainties related to the inundation modelling approach and to the chosen vulnerability functions. These uncertainties are considered in the model experiment by varying the setup of the submodules for flood modelling and loss modelling.

### 2.1 Probable maximum precipitation and probable maximum discharge

The probable maximum precipitation PMP for the whole catchment was estimated using the guidelines of World

Meteorological Organization (2009). The method for distributing the PMP in space and time is based on a Monte Carlo approach proposed by Felder and Weingartner (2016). This approach aims at identifying a PMP pattern leading to the PMF





by testing a high number of randomly generated spatio-temporal patterns under consideration of physical plausibility criteria. To consider the spatio-temporal patterns of precipitation in the river basin, the same amount of areal precipitation in the PMP scenario (300 mm for a 72 h event over 3000 km$^2$) was distributed in different spatio-temporal patterns across the entire river basin in a Monte-Carlo-simulation framework after Felder and Weingartner (2016). We focused on a precipitation event

lasting three days, since this time span corresponds to the typical event duration within the river basin and leads to the highest floods. The PMP scenarios are assumed to occur during the summer season with a height of the freezing level above the maximal altitudes. This means that snowfall is not considered. In a first step, a random temporal distribution of the total precipitation for the chosen duration was generated. In a second step, the temporal pattern of the rainfall was distributed spatially in three meteorological regions, and in five sub-catchments within each meteorological region. The sub-catchments

and the meteorological regions were defined to consider the relatively independent behaviour of specific parts of the catchment, e.g. lowlands and mountainous regions, in terms of precipitation amount and intensity. The randomly created precipitation pattern was checked against the spatial dependencies (affiliation of sub-catchments to wider meteorological regions to fulfil a spatial consistency within neighbouring catchments, for further details see Felder and Weingartner (2016). From a set of 10$^6$ Monte-Carlo simulations with a simplified but computationally efficient hydrological model based on unit

hydrographs, we selected 150 scenarios with the highest discharge at the basin outlet in Bern. The number of scenarios is chosen to allow analysing the variability of PMP patterns but at the same time allowing to be computationally feasible. These precipitation scenarios are then used as inputs for the detailed rainfall-runoff model, which is set up for each tributary and delivers the input hydrographs for the hydrodynamic model. For the rainfall-runoff modelling, we used the hydrological model PREVAH (Viviroli et al., 2009b), which is a deterministic, semi-distributed, HRU-based model, where the

hydrological response units are directly routed to the catchment outlet. The model is set up for 15 sub-catchments that are located within the Aare river basin upstream of Bern using an hourly time steps. The calibration and validation of the hydrological model is described in (Felder et al., 2017). The output of the hydrologic model of each sub-catchment is used as an upper boundary condition for the hydrodynamic model, in this case the 1D hydrodynamic model BASEMENT-ETH (Vetsch et al., 2017) that accounts for the retention effects of lakes and floodplains. The model is based on the continuity

equation and solves the Saint-Venant equations for unsteady one-dimensional flow. Lakes and their outlet weirs are considered in the hydrodynamic model. Here, we considered only the discharge from the lakes with maximal open weirs. No lake or reservoir regulation is considered, since lake regulation can be assumed as not relevant in case of extreme floods. The hydrologic and the hydrodynamic models were calibrated and validated separately, and then again together in the coupled version. The hydrologic model was calibrated with all available gauged observation data at the outflow of 8 out of the 15

sub-catchments. The models for the ungauged sub-catchments were regionalized by applying the parameter regionalization method proposed by Viviroli et al. (2009a). The 1D hydrodynamic model was calibrated by empirically adjusting the friction coefficients in the river channels with particular regard to the water surface elevation in the main channel at peak discharge. However, the coupled hydrologic hydraulic model was validated against the observation at the catchment outlet. In the



validation period 2011-2014, the coupled hydrologic-hydraulic model has a NSE value of 0.85 (Nash-Sutcliffe-Efficiency; Nash and Sutcliffe, 1970), and a KGE value of 0.85 (Kling-Gupta-Efficiency; Gupta et al., 2009; Kling et al., 2012).

## 2.2 Inundation modelling

The coupled simulations of the 150 rainfall patterns provide the basis for the inundation modelling. We nested the 2D
inundation models into the 1D hydrodynamic model (see a schema of the approach in Fig. 1). The 1D hydrodynamic model computes the level of the lakes and the outflow from the lakes. The hydrographs from the 1D hydrodynamic model and the hydrographs computed by the hydrological model that are directly flowing into the floodplains were used as upper boundary conditions for the 2D flood inundation modelling. Minor tributaries are neglected as upper boundary condition. However, the outflows from their catchments are taken into account by aggregating all minor tributaries to sub-catchment level. The
spatial setup of the model experiment, as well as the interfaces between the hydrological model and the floodplains modelled in 2D, are shown in Fig. 2.

We used the LISFLOOD-FP model for the 2D inundation simulation and as a basis for flood loss modelling. The model and it's validation is described by Bates and Roo (2000), Bates et al. (2010), Neal et al. (2009), Neal et al. (2011), and Neal et al. (2012a). The model was set up with a subgrid representation of the channel and a spatial resolution on the floodplain of 50
m. The digital terrain model (DTM) was upscaled from a Lidar DTM with a high spatial resolution (0.5 m). The basis of this terrain data is a digital terrain model (DTM) provided by the Canton of Bern. This terrain model was created from Lidar-measurements collected in 2014 and 2015 with a resolution of about 4 points per m$^2$. The Lidar data has been processed by the data provider to create a raster DTM with a cell size of 0.5 m. The buildings and the most important hydraulic structures in the main rivers (main bridges) were removed by this process. We corrected this raw raster model by a) manually
eliminating the remaining hydraulic obstacles in the river reaches, b) correcting the height of the riverbanks in the Aare and Guerbe river reaches on the basis of dGPS measurements along the riverbanks, and c) interpolating the altitudes of the raster cells of the river bed on the basis of surveyed bathymetric cross sections provided by the Federal Office for the Environment (BAFU). The result is a DTM with a spatial resolution of 0.5 m and the above mentioned corrections. This hydraulically correct DTM provides the basis for the aggregation at coarser spatial resolution for the flood inundation models.

The subgrid channel module requires the heights of the river bed and of the lateral dams, the river width, and the shape of the river bed. This data was computed at high resolution and aggregated onto the target resolution of the inundation model by conserving the cross-sectional area of the river channel from the high-resolution terrain model.

The 2D hydrodynamic model was calibrated in terms of reproducing the stage-discharge relationships at the gauging stations and the known channel capacity along the river reaches. The model was validated on the basis of documented flooding. The
fit of the inundation model (after (Bates and Roo, 2000) computed on the basis of observed discharges of the flood event in August 2005 and a comparison between modelled and observed inundation extents ranges between 0.5 and 0.9, depending on the floodplain. The lower values can be explained by dam breaks that occurred in reality but are not considered in the





model, or by recent changes in the river geometry since the last flood event (implementation of new flood defence measures).

In addition to the 2D inundation model, we elaborated inundation maps from the 1D hydrodynamic simulations. We constructed water surface elevation (wse) maps by interpolating the *wse* values at the cross sections of the 1D model. The

projection of these *wse* maps onto the digital terrain model (spatial resolution of 10 m) and the comparison with the DTM subsequently leads to a map of flow depths.

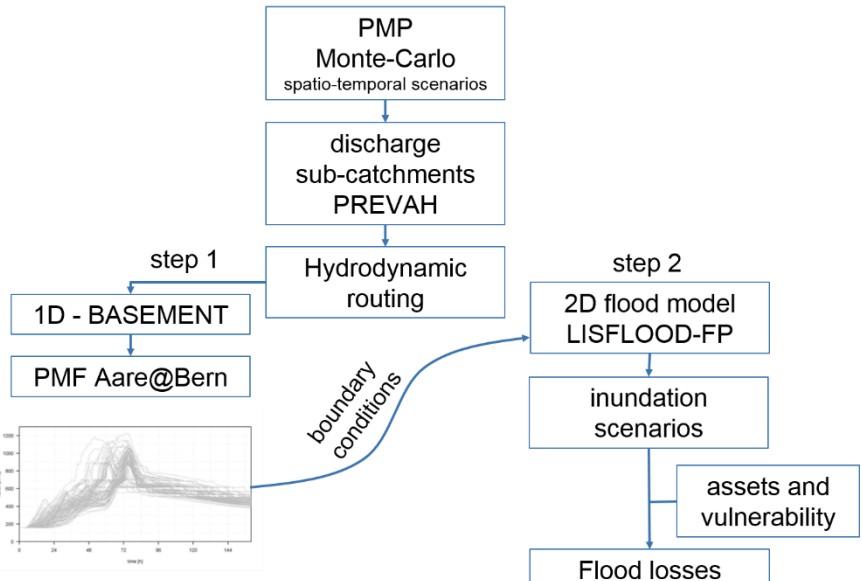

**Figure 1: Schematic of the nesting approach. The 2D flood inundation models and the loss models are nested in a 1D routing**
**model.**

**2.3 Flood loss modelling**

In this study, we focused on structural damage to buildings (residential, public and industrial buildings) without considering losses to mobile assets, building contents and infrastructure. The flood loss module of this model chain consists of a dataset

of buildings similar to that described in Röthlisberger et al., (2017) and Fuchs et al. (2015). Each building is represented by a polygon and is classified by type, functionality, construction period, volume, reconstruction costs, and number of residents. Furthermore, we delineated the height of the ground floor above sea level of each building on the basis of a Lidar terrain model with sub-meter resolution.

The resulting flow depths (*fdm*) and water surface elevations (*wse*) from the hydrodynamic module were attributed to each

building (exposure analysis) and used for deriving the object-specific degree of loss from the vulnerability functions and



consequently for the estimation of object-specific losses. The flow depth was attributed to the building following two different approaches. The first approach is a direct attribution of the flow depth from the *fdm* maps to each building. The second approach is an indirect attribution where the flow depth at each building results from the difference between the *wse* from the *wse* raster of the flood simulation and the minimum ground floor level of the building. The flow depth was used to calculate the degree of loss on the basis of a vulnerability function. The degree of loss resulting from the flow depth and the vulnerability function was subsequently multiplied with the reconstruction value of the building. This results in the expected loss to the building structure. The object-specific losses were subsequently summed to give the cumulative losses of a simulated precipitation scenario.

Five vulnerability functions were considered in the damage calculation procedure. We used the functions of Totschnig et al. (2011) (V1), Papathoma-Köhle et al. (2015) (V2), Hydrotec (2001) (V3) as cited in Merz and Thieken (2009), Jonkman et al. (2008) (V4), and Dutta et al. (2003) (V5). We used different vulnerability functions because there is no regionally adopted and validated vulnerability function available for Switzerland, and because we aimed explicitly at exploring the range of uncertainties related to the choice of the function and its relevance for the total uncertainties in the outcomes. A direct validation of the vulnerability functions was not possible because of a lack of loss data at the level of single objects due to privacy restrictions.

### 2.4 Benchmarking against other uncertainties

The effects of variability in probable maximum precipitation patterns on flood losses are compared with other uncertainties. The comparison was made by following the parallel models approach first presented by Visser et al. (2000) for the example of climate simulations. Merz and Thieken (2009) adopted this approach for the identification of principal uncertainty sources in flood risk calculations. In summary, this approach computes a number of model runs with varying input parameters. In a first step, the minimum and maximum values of all simulation outcomes (flood losses in financial units in this study) were extracted. The difference between both is defined as the maximum uncertainty range (MUR). In a second step, the uncertainty range ($UR_{sub}$) of a specific subset of model runs was computed. The subsets from all model runs can be defined by specific criteria, e.g. a subset of all model runs with the same flood model or a subset of model runs using the same vulnerability function. The uncertainty range of this subset is given by the difference between the minimum and maximum values of all simulation outcomes of this specific subset. Third, the reduced uncertainty range (RUR) was computed according to eq. (1). This indicator describes the relative contribution of the subset to the total uncertainty range.

$$RUR = \frac{(MUR - UR_{sub})}{MUR} * 100\% \tag{1}$$

A high value of RUR means that the subset contributes significantly to the total uncertainty range. Alternatively, a small value of RUR (RUR $\ll$ 100 %) indicates that the subset has a reduced effect on the overall uncertainty (Visser et al., 2000). In the model experiment for this study, we analysed the relative contribution of a) the spatio-temporal rainfall pattern, b) the choice of the inundation model and the exposure analysis approach, and c) the choice of the vulnerability function. Hence,




we followed a hierarchical approach for the selection of the subsets. For assessing the contribution of the spatio-temporal rainfall pattern to the overall uncertainty, we analysed 150 rainfall scenarios (hierarchical level 1 - precipitation). For each of these rainfall scenarios, the losses were computed with two different flood inundation models (LISFLOOD-FP and BASEMENT-1D) in combination with two different exposure modelling approaches (flow depth *fdm* and water surface elevation *wse*) (hierarchical level 2 – flood model) and five different vulnerability functions identified previously (hierarchical level 3 – vulnerability). For each PMP scenario, 20 loss estimations were computed (four flood models times five vulnerability functions). Overall, the whole ensemble amounts to 3000 model runs (i.e. flood loss estimations). The RUR values were computed on the basis of subsets selected by the hierarchical levels representing the uncertainty factors considered in this analysis.

## 3 Study area

We set up the flood inundation models for the main valley of the Aare river basin upstream of Bern, Switzerland. The catchment elevation ranges from 500 to 4200 m a.s.l., with a mean elevation of 1600 m a.s.l. The southern part of the river basin consists of relatively high alpine mountains. Several alpine peaks within this area exceed 4000 m a.s.l., and parts of it are glaciated (8% of the total catchment area). The main valley of the Aare River basin consists of a relatively flat floodplain with two lakes, where widespread flooding can occur. The lakes are natural but artificially managed, and are oriented along an approximately east-west axis in the lowland part of the catchment. The study area covers 3000 km$^2$, and the following main river reaches are considered in the model chain (see Fig. 2):

(1) Hasliaare river, from Meiringen to Lake Brienz (floodplain: 15 km$^2$, contributing area: 451 km$^2$)

(2) Lake Brienz static inundation model (lake area: 31 km2, contributing area: 1138 km$^2$)

(3) Interlaken, area between Lake Brienz and Lake Thun and the fan of the Lütschine River (floodplain: 28 km$^2$)

(4) Lake Thun static inundation model  (lake area: 50 km2, contributing area: 2450 km$^2$)

(5) Thun (floodplain: 8 km$^2$)

(6) Aare River reach between Thun and Bern (floodplain: 42 km$^2$)

(7) Gürbe River reach between Burgistein and Belp (floodplain: 15 km2, contributing area: 116 km$^2$)





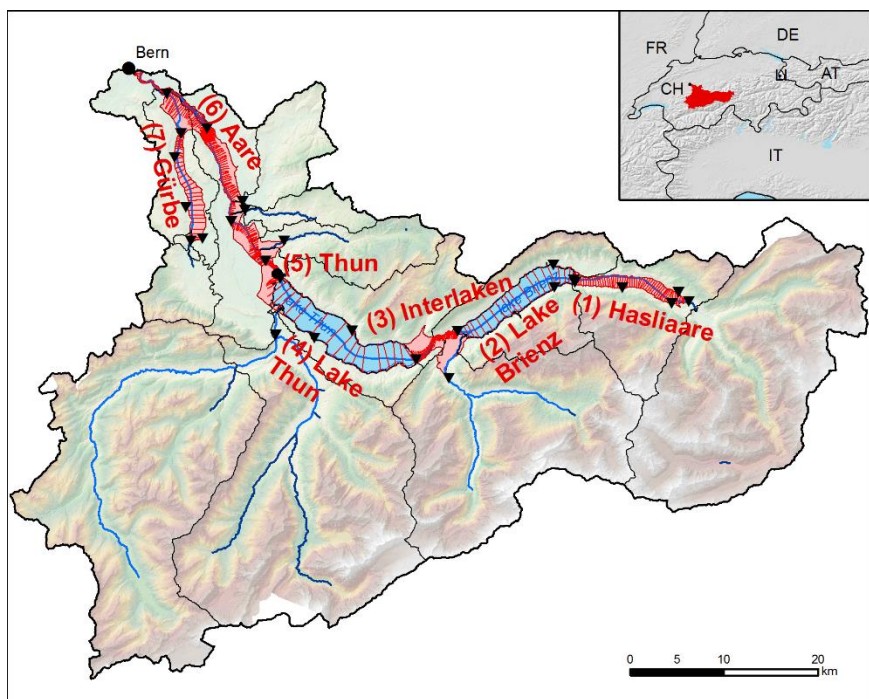

**Figure 2: The Aare river catchment upstream of Bern, Switzerland. The sub-catchments of the hydrological model are divided by black lines. The black triangles indicate the coupling points between the hydrologic and the hydrodynamic model. The 1D routing model covers all floodplains (red lines) and the lakes (blue). The floodplains that are covered by the individual 2D inundation models nested into the 1D routing model are marked and labelled in red.**

## 4 Results

The main results of the coupled model simulations are the discharges at the outlet of each of the sub-catchments, the discharge at the outlet of the Aare river basin at Bern, and the flood losses for 150 PMP simulations. Figure 3 shows the hydrographs of the 150 PMP scenarios at the outlet of the river basin in Bern. The outflow from the river basin varies remarkably in peak discharge and time to peak. The peak discharges for each ensemble member were in the range 906 to 1296 m³/s. Thus, the highest peak discharge is 43% higher than the lowest in the selected set of scenarios.

The flood inundation modelling resulted in a set of flood maps representing the 150 PMP scenarios. The overlay of these flood maps leads to a probabilistic inundation extent map. Each inundation map is treated as equally weighted in the probabilistic map. This map represents the probability that a model grid cell is flooded in one PMP scenarios. An extract of this map is shown in figure 4. The map shows that not all of the PMP scenarios lead to flooding of the same areas. Thus, despite the narrow framing of floodplains in mountainous areas by topography, a high variability in flood extent can be observed. The discharge in the Lütschine River at Interlaken and the lake levels of both lakes have the strongest influence on the inundation probability map. In particular, the level of Lake Thun determines a remarkable portion of the flooded area.





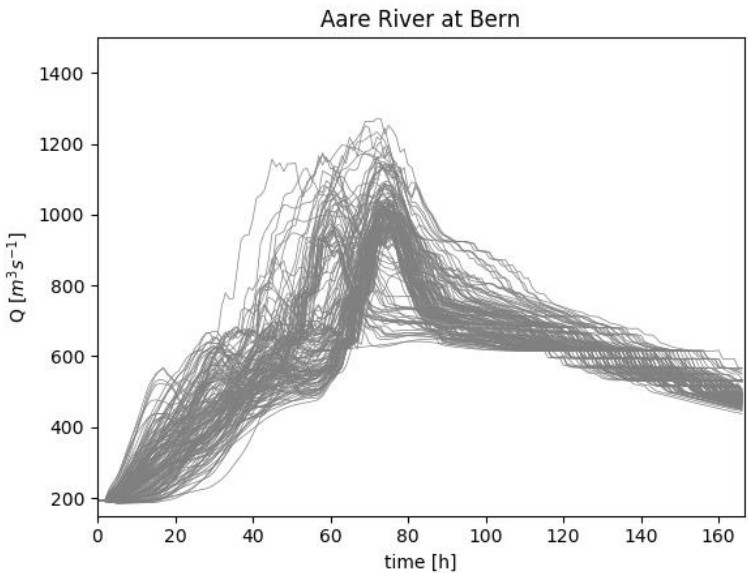

**Figure 3: Hydrographs at the outlet of the Aare River basin in Bern resulting from the coupled hydrologic-hydrodynamic modelling of the 150 PMP scenarios.**

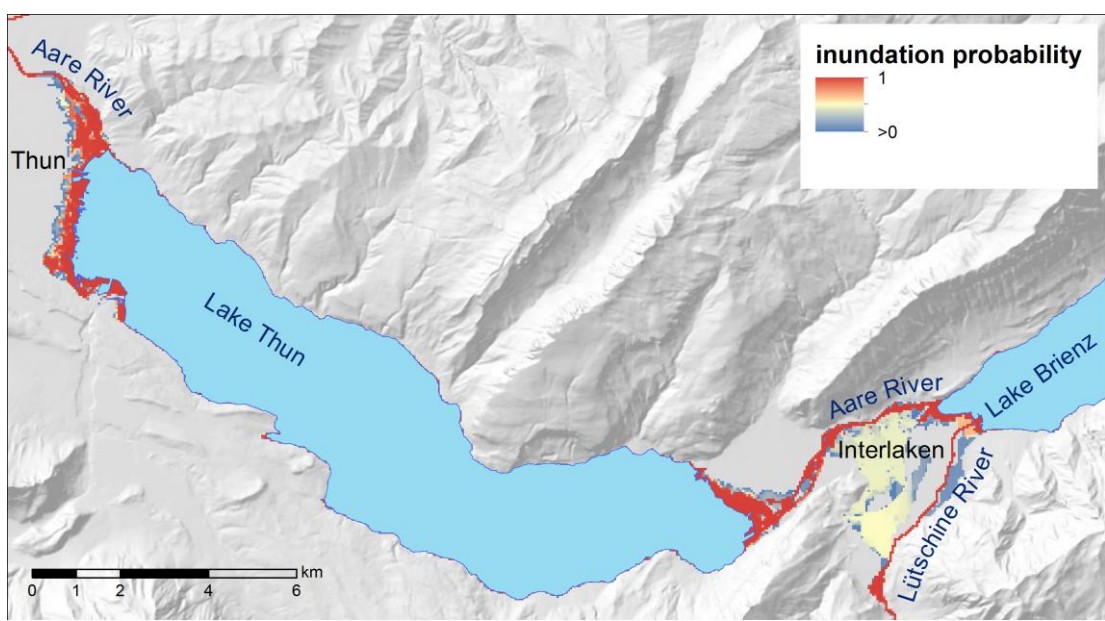

**Figure 4: Detailed example of the probabilistic flood map for the floodplains of Thun and Interlaken. Predicted flood inundation extents can change significantly depending on the specific spatial properties of a few of the PMP scenarios and hence have lower mapped inundation probabilities.**




Depending on the chosen approach for inundation modelling and exposure analysis, the number of affected buildings varies remarkably. At minimum 2423 buildings and at maximum 4667 buildings are affected across the whole domain (not shown). The high variability between the PMP patterns is also shown by the number of exposed residents (figure 5).

The flood simulation mapped outputs (flow depth maps and water surface elevation maps) were used separately to calculate the flood losses at single building level. Subsequently, the flood losses at building scale were aggregated at a catchment

level. Figure 6 shows the distribution of the aggregated flood losses. It is shown that – depending on the model ensemble member – the losses vary between 0.06 and 2.2 billion Swiss Francs (CHF). Thus, the losses are remarkably influenced by all the experimental uncertainty factors previously discussed in the modelling chain. However, even if the effect of the vulnerability function and the choice of the exposure analysis approach are not considered, the losses still vary markedly depending on the PMP scenario. Maximum losses are still approximately 3-5 times the minimum losses for some of the

vulnerability functions. The vulnerability function V4  (Jonkman et al., 2008) results in the lowest losses. This function was calibrated for lowland floodplains and thus has generally lower degrees of loss. However, this vulnerability function might be more representative for the areas affected by lake flooding than the others.

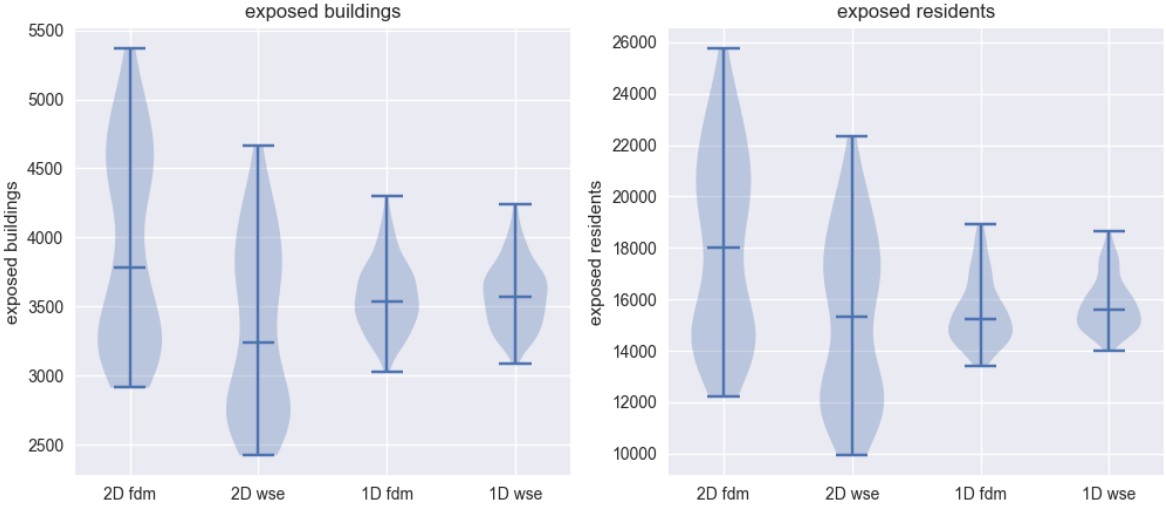

**Figure 5: Exposed buildings (left) and residents (right) aggregated at river basin level. Flood losses aggregated at river basin level.**
**The variation between the PMP scenarios is shown in the y-axis, whereas the x-axis shows the variability inherent to the choice of the flood model (2D: LISFLOOD-FP, 1D: BASEMENT-1D) in combination with the approach for attributing flow depths to the buildings (*fdm*: flow depths are calculated on the basis of flow depths maps, *wse*: flow depths are calculated on the basis of water surface elevation maps and the object-specific ground floor level).**



The benchmark against other uncertainties such as the flood modelling in combination with the exposure analysis approach and the vulnerability functions shows that all the uncertainties considered in the model experiment contribute significantly to the total uncertainty range. Each member of the ensemble runs represents a rainfall pattern and a resulting flood loss computed on a basis of a combination of a specific flood model with a specific loss model. The difference between the

ensemble member with the absolute minimum and the member with the absolute maximum of flood losses represents the total uncertainty range MUR. The total number of runs was divided into subsets that represents in each case the uncertainty range of a specific combination of the variables. The difference between the member with the absolute minimum of this subset and the member with the absolute maximum of this subset represents the reduced uncertainty range $UR_{sub}$. Consequently, the reduced uncertainty range RUR is computed after eq. (1). The reduced uncertainty range RUR of all

subsets ranges between 14 and 92 % of the total uncertainty range MUR. The reduced uncertainty range of the subset of ensemble members considering only the variability in rainfall scenarios lies between 42 and 92% with a median of 72%. Hence, the highest RUR of all subsets is dominated by the subsets regarding the variability in probable maximum precipitation pattern. Figure 7 shows the comparison between the RUR values of the subsets in which the variability of one of the three considered uncertainty factors was analysed. This analysis makes evident that the rainfall pattern contributes

most to the total uncertainty range.

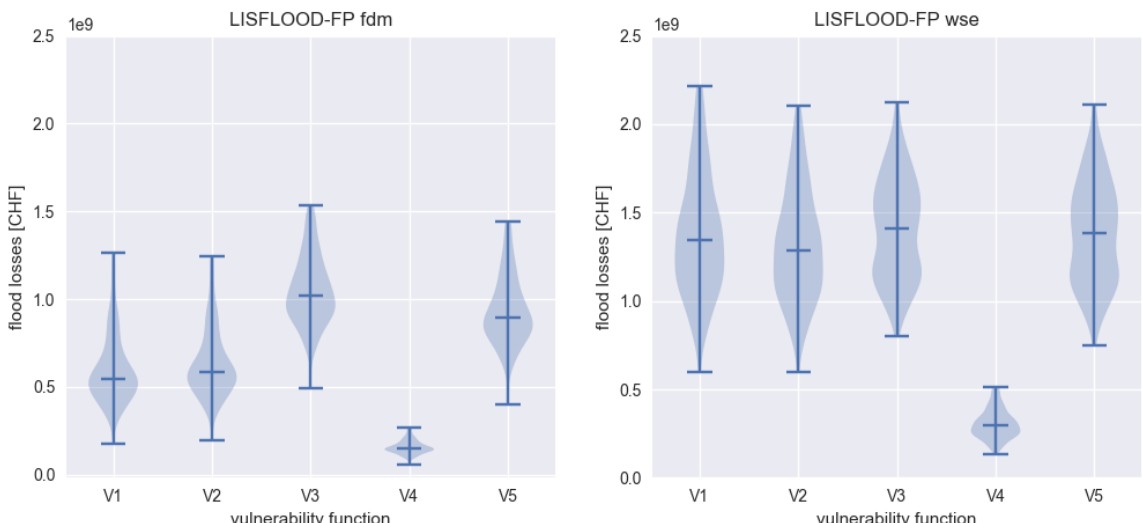

**Figure 6: Flood losses aggregated at river basin level. The variation between the PMP scenarios is shown in the y-axis, whereas the x-axis shows the variability inherent to the vulnerability functions. The diagram at the left shows flood losses that are calculated based on the flow depths as modelled by LISFLOOD-FP, the diagram at the right shows the flood losses that are calculated based on the water surface elevation and the object-specific ground floor level.**





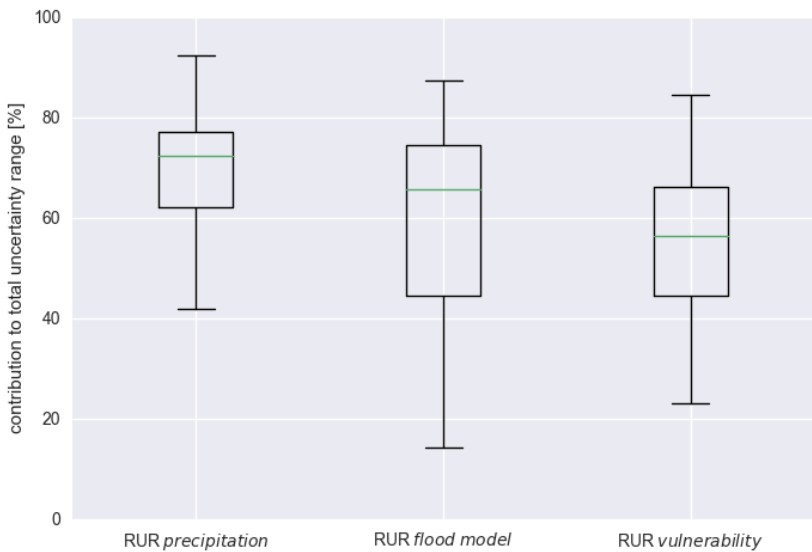

**Figure 7: Reduced uncertainty ranges RUR of the subsets of model runs representing the three main uncertainty sources.**

In figure 8 (left column), we plotted the results of all model outcomes with focus on the 2D inundation model in terms of

exposed number of buildings and persons, and in terms of flood losses against the peak discharge of the respective precipitation pattern at the catchment outlet. The hypothesis that the flood losses increase with peak discharge at the outlet of the river basin can be verified in the sense that there is a significant correlation. This relationship is weaker for exposed buildings and residents than for the flood losses. However, the rainfall scenario leading to the highest peak discharge at the basin outlet does not correspond with the highest flood losses. Instead, the flood losses are more correlated with high lake

levels in Lake Thun (see figure 8, right column). The correlation between flood losses and the level of Lake Thun (Spearman's rank correlation coefficient ranges from 0.54 to 0.94, depending on the flood model and the vulnerability function) is stronger than between losses and the peak discharge at the catchment outlet (Spearman's rank correlation coefficient ranging from 0.43 and 0.71). Thus, in the Aare River basin, the level of Lake Thun is a more relevant proxy indicator for the amount of flood losses in the whole river basin than the peak discharge at the outlet of the river basin (i.e.

the so-called PMF of the river basin). This can be explained by the local situation of the city of Thun where the density of the building stock is very high along the shoreline of Lake Thun and along the Aare River. The major area of the Aare river basin contributes to the lakes. Only 20% of the catchment area is located downstream of Lake Thun. Although the area of Lake Thun covers only about 2% of its contributing area, this means that the river basin exhibit relevant retentions areas that attenuates the outflow from the river basin and thus the PMF. Vice versa, this retention effect increases flood losses because

a relevant number of buildings is located in neighbourhood of the lake shorelines. Likewise, not all of the PMP scenarios lead to flooding by the Lütschine river in Interlaken. As shown in figure 4, the floodplain of this river is flooded only in a





minority of the ensemble runs. Whether this floodplain is flooded or not accounts for up to 1500 exposed buildings and therefore up to one third of the total number of maximally exposed buildings in the whole river basin. Thus, the highest loss of all simulated scenarios is related to a combination of a high lake level in Thun with a high river discharge of the Lütschine river. This documents that the maximum loss depends on both, the spatio-temporal pattern of the rainfall and the internal

organization of the river basin in terms of the spatial distribution of the values at risk (i.e. exposure) within the floodplains.

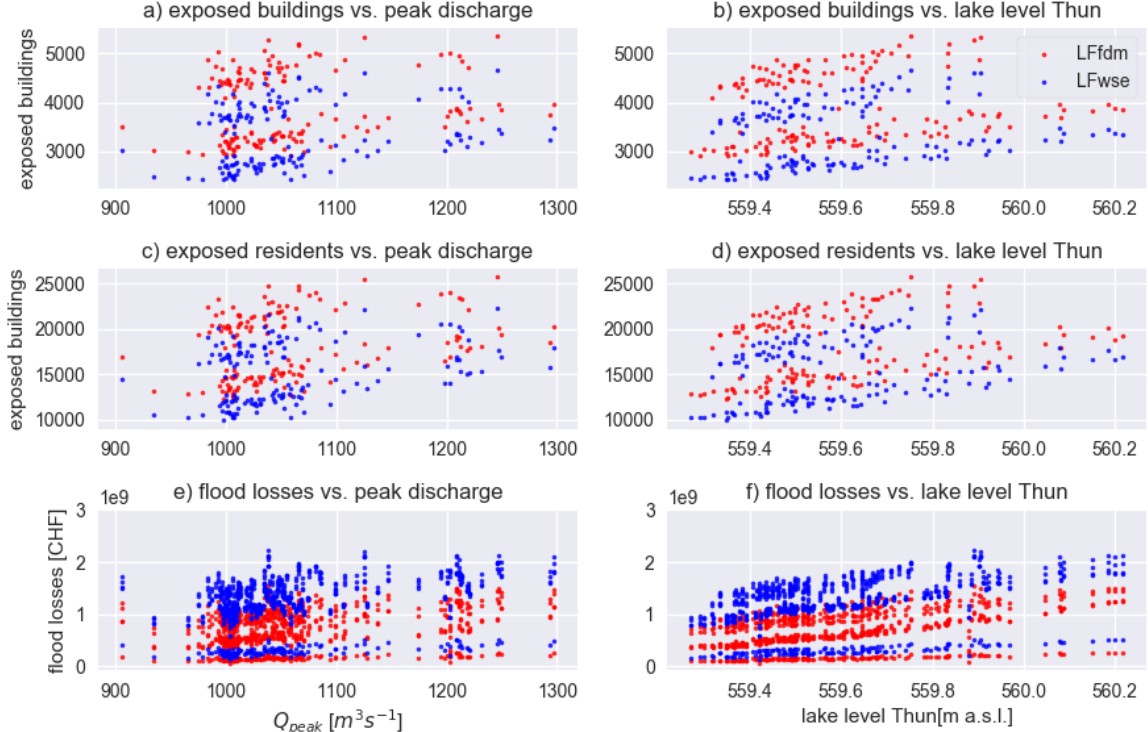

**Figure 8: This figure shows the aggregated flood losses for the 150 PMP scenarios. The red dots show the exposed entities and losses that are computed based on the flow depths, the blue dots show the exposed entities and losses that are computed based on water surface elevation. The figures in the last row show the losses resulting from all vulnerability functions.**

**5 Discussion and conclusions**

In this study, we set up a coupled component model for estimating flood losses of extreme flood events in a complex mountainous river basin. On the basis of a Monte Carlo approach, we computed an ensemble of extreme flood events for different precipitation patterns of a three-day probable maximum precipitation scenario. With this model experiment, we analysed the effects of the spatial distribution of the rainfall within a mountainous river basin on flood losses. Furthermore,

we benchmarked these effects with other uncertainties in flood loss modelling.

The model experiment showed that the spatial variability of the precipitation patterns within a river basin with a complex topography is one of the main contributions to the overall uncertainties in flood loss modelling at river basin scale. The PMP





pattern determines the magnitude and timing of the flood peaks coming from the sub-catchments and flowing through the floodplains along the main valleys and the lakes. Thus, the rainfall pattern could lead to a superimposition of flood waves as described by the model experiments of Neal et al. (2013) and Pattison et al. (2014). In addition to the superimposition of flood waves, it is shown that lake levels, as a proxy for the water volumes coming from different sub-catchments, are also

relevant for the determination of flood losses. This complements the findings of Sampson et al. (2014) on the impacts of precipitation variability on insurance loss estimates. With the present study, we extended the Sampson *et al.* study with a focus on urban environments with a focus on complex mountainous river basins.

Furthermore, the model experiment showed that the peak flow coming from a single sub-catchment can be responsible for a relevant share of the total sum of exposed buildings and flood losses. Thus, the physical variability of the river basin is

coupled with the topological situation of the main settlements within the floodplains, i.e. the spatial pattern of exposure. The inundation probability maps and the variability in flood losses show that two floodplains are mainly responsible for a high amount of flood losses. This documents that flood losses depend on both, the spatio-temporal pattern of the rainfall and the internal organization of the river basin in terms of the spatial distribution or aggregation of the values at risk within the floodplains. Moreover, the spatial setup of the values at risk within the floodplains leads to its specific sensitivity to flood

magnitude and lake level. However, these specific sensitivities of the single floodplains together with the variability in rainfall pattern lead to a specific sensitivity of the whole river basin to a certain pattern of rainfall. This behaviour has to be analysed and generalized in further studies and considered in the estimation of probable maximum flood losses.

Despite the topographical confinement of the floodplains by the mountain hillslopes, the flooded areas vary markedly with different rainfall patterns. Thus, the probabilistic map shows a high spatial variability, caused by a few of the PMP scenarios

significantly increasing inundation areas. Hence, also the flow depths at single buildings, and consequently the total flood losses, vary remarkably with rainfall scenario. This case study in a mountainous environment and in an environment with remarkable retention capacities due to the presence of lakes may even lead to an attenuated illustration of this effect. These retention effects attenuate the PMF on one side but control the flood losses on the other side if settlements are located alongside the lakes. Though, in mountain areas without lakes, the effects of spatio-temporal variability in precipitation

patterns on flood losses may be even more accentuated. However, a modelling approach is needed to analyse these effects as stated by Emmanuel et al. (2015).

Nevertheless, the other uncertainties considered in this study, i.e. the role of the flood model, the exposure assessment approach and the vulnerability functions, are also contributing markedly to the total uncertainty range. This is in line with the findings of other studies (Jongman et al., 2012; Moel and Aerts, 2011). Consequently, these uncertainties also have to be

taken into account in a portfolio analysis or in the analysis of probable maximum flood losses.

In summary, we conclude that the analysis of a broader set of extreme floods with different precipitation patterns leads to more a comprehensive view of flood losses in a river basin compared to standard deterministic PMP/PMF methods. The spatio-temporal characteristics of rainfall patterns must be considered in complex mountainous river basins. Moreover, the analysis of the probable maximum flood losses in a river basin should consider the systemic vulnerability of the floodplains



or the behaviour of floodplains as human-water systems as stated by Di Baldassarre et al. (2013, 2014). This involves the identification of key locations of exposure that contribute most to the overall flood losses. Probabilistic inundation maps provide a first overview of key locations of flooded areas with a high sensitivity against the rainfall pattern. Furthermore, it is shown that the presented model experiment provides a valuable instrument for the consideration of all components in the analysis of the variability of rainfall patterns to flood losses in a river basin, from hazard to exposure and vulnerability.

In future research, an inverse modelling approach may be followed by searching the worst case precipitation pattern leading to the worst case flood losses on the basis of the system characteristics of the river basin (sensitivities of floodplains and spatial setup of the river system). The calculation of the maximum expectable flood losses in a river basin should not be based exclusively on the PMF. In contrast to the initial hypothesis, we observed that other catchment characteristics in addition to the PMF could remarkably influence the flood losses. Consequently, in complex river basins it is recommended to analyse the sensitivity of the most relevant floodplains before analysing the probable maximum flood losses.

*Code and Data Availability*: Data of cross section surveys were provided the Federal Office for the Environment. The Lidar terrain model was provided by the Canton of Bern. Basic GIS-data were provided by the Federal Office for TopographySwissTopo, the residential register was provided by the Federal Office for Statistics. The data about values at risk are restricted by privacy regulations. All other data produced in this study and the codes for the model experiment are available from the leading author on request. The inundation model LISFLOOD-FP is available at http://www.bristol.ac.uk/geography/research/hydrology/models/lisflood/.

*Author contributions*: The study was designed by A. Zischg, J. Freer, P. Bates, and R. Weingartner. The hydrologic model was set up and run by G. Felder. Model coupling and the set up of the hydraulic model were done by A. Zischg, N. Quinn, G. Coxon, and J. Neal. The loss model was developed by A. Zischg. The analyses were performed by A. Zischg with the support of all co-authors. The manuscript was prepared by A. Zischg with the contribution of all co-authors.

*Competing interests*: The authors state no conflict of interest.

*Acknowledgments*: The work was partially funded by the Swiss National Foundation (grant number IZK0Z2_170478/1), by the Swiss Mobiliar, by NERC grant SINATRA (Susceptibility of catchments to INTense RAinfall and flooding, grant number NE/K008781/1), and by NERC grant MaRIUS (Managing the Risks, Impacts and Uncertainties of droughts and water Scarcity, grant number NE/L010399/1). We thank the Federal Government of Switzerland and the Canton of Bern for delivering the data.





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
