# Peer review of "Effects of variability in probable maximum precipitation patterns on flood losses"

_Hydrology and Earth System Sciences, 2017_

## Referee Comment (RC1) · Anonymous Referee #1 · 5 Mar 2018

This paper presents a model-chain sensitivity analysis for flood loss estimation using the probable maximum flood (PMF) approach. While many authors have studied the sensitivity of flood characteristics to the spatio-temporal distribution of rainfall, the novelty of this paper is to look at the sensitivity of estimated flood losses, which is at the end of the risk assessment chain.

The paper is well written and properly concise. I am not a big fan of the concept of PMF but I understand that, since it is used in practice, research on it is noteworthy. My main comments/suggestions are the following:

- Reading Page 5, lines 7-12, it seems that you assume multiplicative space-time separability for rainfall, meaning that storm movements are not accounted for. Based on my experience, and since you are interested in synchronisation of floods, the fact that the

storm moves over the catchment may have a significant impact on the flood peak at the outlet (and maybe on the flood losses). Given that the models that you use seem to be capable of accounting for it, you should explain why you did not consider this aspect, or to include it in your analysis.

- Analogously, it seems that flow velocities are not accounted for in the flood loss modelling part. You should motivate why you believe this is not necessary. By the way, may the fact that, in your results, flood losses correlate more with the lake levels than the river discharges at the outlet be due to considering water levels only and not velocities in the loss model?

- I know that, in different research areas, the definition of "uncertainty analysis" and "sensitivity analysis" are different and sometimes exchangeable. In my view, what you do in this paper is a sensitivity analysis on the assumptions you make (which of course are affected by uncertainty). To me, an uncertainty analysis requires to model statistically the uncertainties in the assumptions made, i.e., through probability distributions, and to propagate these distributions through the model chain to the output distributions. You somehow do this for the rainfall spatial patterns, using the Monte-Carlo approach, but not for other choices such as for the vulnerability functions or the parameters of the models. It would be useful to state, for example after line 10 at page 4, how you define "sensitivity analysis" and "uncertainty analysis" and what you are going to show in the paper.

Additional detailed comments:

Page 4, line 24: what does "assessing uncertainties in the model output" mean?

Page 8, eq. (1) (and Figure 7 at page 14): I guess the sum of RUR for the different factors does not equal 100% (i.e., global sensitivity is normally not the sum of local sensitivities). Am I right? If so, RUR does not quantify the relative contribution of the different components to the total uncertainty but it is somehow related to it (it would be interesting to understand how it is).

Page 10, line 13 (and legend of Fig. 4): how can the flood inundation maps be "probabilistic" if the probability of the PMP is unknown? I guess you mean the maps represent estimates of the conditional probability of inundation, conditional to the PMF to have occurred.

Page 13, line 3: do you really think that the total uncertainty range has been quantified in the study? It is to me the range of sensitivity to the different assumptions made.

Page 15, line 17: I would say that, given the range of assumptions made here, the sensitivity of flood losses to the variability of spatial distribution of rainfall is larger than for other factors.

Page 16, line 2: could the superimposition of flood waves be quantified for your scenarios?

---

## Referee Comment (RC2) · Anonymous Referee #2 · 9 Mar 2018

**Comments on manuscript titled: "Effects of variability in probable maximum precipitation patterns on flood losses"**

Authors: Andreas Paul Zischg, Guido Felder, Rolf Weingartner, Niall Quinn, Gemma Coxon, Jeffrey Neal, Jim Freer, Paul Bates
Article type: research paper
Note: all the comments are based on the electronic document 'hess-2017-758.pdf'

I think that the manuscript presents an important piece of research work which helps to enhance the understanding of several effects on flood loss estimation using the model chain approach which consisting of five components starting with a precipitation module based on PMF approach and ending with a damage module.

The paper is in general well-structured and presented in a quite clear manner. Readers, at least like me, feel comfortable to follow and understand the story of the study reading the excellent introduction chapter. However, going through the remaining chapters I have several points which need to be addressed and clarified by the authors:

- P6, L4-11: Pls explain what "nested" means and why we need this in this context? P6, L6-8: how about the downstream boundary condition? Additionally, what shown in Figure 2 is quite confusing: why do we need the 2D model when we have already 1D model covering the whole flood plain? How to define coupling points (denoted as black triangles in the figure).

- Figure 1 is not clear and should be improved: (1) at the curved arrow line "boundary conditions", where does the bunch of hydrographs come from? (2)pls expand the figure for the flood loss component adding info on the use of fdm and wse maps and the five vulnerability functions (V1…V5) because they are used intensively in subsequent plots and discussions.

- Have the exposed buildings/residents corresponding to 1D fdm, 1D wse (Figure 5 and Figure 6) been used for any loss calculation? If not, Pls remove them from the two figures. Additionally, can the authors explain that the distribution of exposed buildings seems to be bimodal in the cases of 2D but unimodal in the cases of 1D?

- P12 L6: "At minimum 2423 … maximum 4667 buildings….". Are these numbers correct? I see at least in the 2D fdm case, the maximum value exceeds 5300 buildings.

- I may miss some import points when reading the manuscript! The exposed buildings are higher in 2D fdm compared to those of 2D wse (Figure 5), could the authors help explain why their corresponding losses the opposite (Figure 7 and Figure 8). To this end, I think I would be clearer to explain 'fdm' and 'wse' in a better way (meaning P7L19 – P8L8 is not clear enough!).

- P1L15 and elsewhere: "other uncertainties". There are several kind of uncertainties of which some can be quantified the other cannot. Therefore, the use of "other uncertainties" in this sentence is not recommended even though I understand what the authors mean!

- Figure 3: To see clearer spatio-temporal pattern, it would be better to add hydrographs at the outlet of some sub-catchments.

- P6 L4: delete "from the wse"

---

## Author Comment (AC1) · 18 Mar 2018

Dear reviewer, Thank you very much for helpful remarks. Below you find a response to the points made.

-Concept of PMF:

We agree that the concept of the PMF is debatable, from several points of view. Hence, we aimed at adding two further aspects to the discussion on methods for stress tests or flood loss analyses at river basin scale, i.e. the role of the spatio-temporal variability of rainfall and the relationship between PMF discharge at catchment outlet and the flood losses.

-Space-time separability and storm movement:

[Figure]

Our concept considers the variability of rainfall of both in space and time. Thus, the storm movement is implicitly considered in our approach. We added plausibility checks for the selection of the stochastically generated rainfall patterns, e.g. the concentration of rainfall in neighbouring catchments in space and time. Furthermore, the storm movement in our case study is influenced by the mountain crests that divide the sub-catchments. Thus, intense precipitation could be concentrated in a few neighbouring catchments. The temporal component of our rainfall distribution model corresponds to the observed ones. Conclusively, we preferred the Monte Carlo approach against a storm movement approach. Full details on this are given by Felder and Weingartner (2016, 2017). We will add more detail about this discussion in the paper.

-Flow velocities:

We do not consider flow velocity because the chosen flood models are not validated against flow velocities in the floodplains. Furthermore, the majority of vulnerability functions does not consider flow velocity. Thus, we focused on comparable vulnerability functions. We will motivate our choice in the paper.

-Uncertainty analysis vs. sensitivity analysis:

Thanks for this advice. We will define the terms uncertainty, sensibility and variability precisely and we will use them in a concerted way throughout the paper.

-Assessing uncertainties in the model output:

We will define what we consider as model output in the paragraph before this sentence to ease readability.

-RUR:

That's right. The sum of RUR for the different factors does not equal 100%. Consequently, the RUR of a subset of models shows the relative role of this uncertainty source to the total uncertainty of all model runs. The RUR is related to the total uncertainty range of all models but is not relative to the RUR of other subsets. The RUR

does not isolate all the contributions of the different components to the total uncertainty but they remain intertwined, except the selected uncertainty factor. However, the RUR values are comparable. We will add more detail on this in the paper and define the relative contribution more precisely.

-Probabilistic inundation maps:

Exactly, the inundation maps are estimates of a conditional probability of inundation, conditional to the rainfall sum in the catchment within a selected time period. Thanks for this hint, we will reformulate this in the paper.

-Total uncertainty range:

In a strict sense, we do not model statistically the uncertainty with probability distributions. However, we quantified the variability related to selected uncertainty factors. In contrast to sensitivity analyses for a single model, we investigate selected uncertainty sources in flood loss assessment in a multi-model framework. Thus, different concepts are combined. However, as stated before we will define precisely the terms variability, uncertainty and sensitivity analysis.

-Superimposition of flood waves:

We can quantify the superimposition of flood waves at every location within the catchment. We will think about how to visualize this superimposition at selected representative locations within the river basin and we will add an additional figure.

---

## Author Comment (AC2) · 18 Mar 2018

Dear reviewer, Thank you very much for the interesting questions and remarks. Below you find a response to the main points.

-Nesting approach:

The 1D hydrodynamic model is routing the water flow from the subcatchments (upper boundary conditions) towards the catchment outlet. This model considers the weirs and lake outlets and thus calculates the the lake level. The 2D model is used for estimating the flow depths in the floodplains required for flood loss analysis. However, one can base loss estimations either on the 1D model only or on the 2D flood model only. Here, we wanted to show the differences in flood loss estimation when based on

1D or 2D, respectively. The simulation of lake retention with a 2D model is computationally demanding. To save computation time, we simulated all scenarios with the 1D model and nested the 2D model into the outcomes of the 1D model at specific locations (boundary conditions). You justifiable asked for the downstream boundary conditions: The nested 2D simulation of all the floodplains except the one with the catchment outlet require a downstream boundary condition. This is given by the output of the 1D model, in our case the hydrographs of the lake levels. However, in the paper we will explain the nesting approach in more detail.

-Definition of the coupling points:

We defined the coupling points following a bottom-up approach: First, we delimited the floodplains for which the flood loss estimation will be valid (system delimitation). Second, we defined the upper boundary conditions of these floodplains. Third, we delimited the upstream catchments for the hydrological model on the base of this coupling points. However, the location of the gauging stations is considered as well, for calibrating and validating the hydrologic model. We will describe this procedure in more detail in the text.

-Figure 1:

We will improve the figure

-exposed buildings/residents corresponding to 1D fdm, 1D wse:

We used the exposed buildings/residents corresponding to 1D fdm and 1D wse for calculating the reduced uncertainty ranges RUR and for assessing the contribution of the model choice to the total uncertainty range in the flood losses. However, we did not show 1D fdm and 1D wse exposure in figure 6. We will add two more subplots in figure 6 for the 1D results. Furthermore, we will explain the distributions in the exposure. This is mostly related to the inundation model in combination with the rainfall pattern. The 2D model considers the exposure of houses at the upper boundary conditions of the
alluvial fans of the tributary Lütschine while the 1D model considers only the inundation of the main river Aare. Thus, the 2D exposure distribution can be bimodal, depending on the flooding by tributaries.

-maximum number of exposed buildings/residents:

Indeed, the maximum number of exposed residents over all simulations is 5371. The value of 4667 is the maximum of the 2D wse simulations only. Sorry, for this inadvertency. The minimum number is exact.

-"other uncertainties":

We will not use this term in the revised version and specify what we mean.

-Spatio-temporal pattern:

We will visualize hydrographs at selected representative locations within the river basin.

———————————————————

---

## Author Comment (AC3) · 29 Mar 2018

In the first reply, we did not answer to one reviewer comment. Here, we add one more point:

RC: I may miss some import points when reading the manuscript! The exposed buildings are higher in 2D fdm compared to those of 2D wse (Figure 5), could the authors help explain why their corresponding losses the opposite (Figure 7 and Figure 8). To this end, I think I would be clearer to explain 'fdm' and 'wse' in a better way (meaning P7L19 – P8L8 is not clear enough!).

AC: This indeed is interesting and it relates to the flow depth attribution from the output of the flood model to the individual buildings. The mean flow depth over all affected

buildings is 0.54 m in the 2D fdm flow depth attribution method and 0.87 m in the 2D wse flow depth attribution method. This results in higher losses although the number of exposed buildings is lower. We will add a description of this in the revised version.
* * *

---

## Author Response (AR1)

**Effects of variability in probable maximum precipitation patterns on flood losses**

Andreas Paul Zischg et al.

Author's response

Dear editor,

First of all, we would like to thank you and the reviewers for the valuable remarks. Herewith, we resubmit a revised version of our manuscript. We addressed all reviewer's comments and updated our author's comments to reflect the changes made in the revised manuscript. Changes in the manuscript are marked in blue.

Reviewer #1

This paper presents a model-chain sensitivity analysis for flood loss estimation using the probable maximum flood (PMF) approach. While many authors have studied the sensitivity of flood characteristics to the spatio-temporal distribution of rainfall, the novelty of this paper is to look at the sensitivity of estimated flood losses, which is at the end of the risk assessment chain.

The paper is well written and properly concise. I am not a big fan of the concept of PMF but I understand that, since it is used in practice, research on it is noteworthy. My main comments/suggestions are the following:

- Reading Page 5, lines 7-12, it seems that you assume multiplicative space-time separability for rainfall, meaning that storm movements are not accounted for. Based on my experience, and since you are interested in synchronisation of floods, the fact that the storm moves over the catchment may have a significant impact on the flood peak at the outlet (and maybe on the flood losses). Given that the models that you use seem to be capable of accounting for it, you should explain why you did not consider this aspect, or to include it in your analysis.

*AC: We extended this paragraph with a more detailed description of the plausibility checks for rainfall pattern generation. In the discussion chapter, we added a discussion about the limitations of this approach (page 5, lines 14-23).*

- Analogously, it seems that flow velocities are not accounted for in the flood loss modelling part. You should motivate why you believe this is not necessary. By the way, may the fact that, in your results, flood losses correlate more with the lake levels than the river discharges at the outlet be due to considering water levels only and not velocities in the loss model?

*AC: We motivated our choice in the paper (page 9 lines 12-14).*

- I know that, in different research areas, the definition of "uncertainty analysis" and "sensitivity analysis" are different and sometimes exchangeable. In my view, what you do in this paper is a sensitivity analysis on the assumptions you make (which of course are affected by uncertainty). To me, an uncertainty analysis requires to model statistically the uncertainties in the assumptions made, i.e., through probability distributions, and to propagate these distributions through the model chain to the output distributions.

You somehow do this for the rainfall spatial patterns, using the Monte-Carlo approach, but not for other choices such as for the vulnerability functions or the parameters of the models. It would be useful to state, for example after line 10 at page 4, how you define "sensitivity analysis" and "uncertainty analysis" and what you are going to show in the paper.

*AC: We agree that uncertainty analyses and sensitivity analyses have bi-products that are interchangeable in some ways. We added definitions of uncertainty analysis and sensitivity analysis after line 10 page 4 and referred to these definitions later on in the method section (page 4, line 28 - page 5, line 2).*

Additional detailed comments:

Page 4, line 24: what does "assessing uncertainties in the model output" mean?

*AC: We defined model input and model output more precisely (page 4,lines 27-28).*

Page 8, eq. (1) (and Figure 7 at page 14): I guess the sum of RUR for the different factors does not equal 100% (i.e., global sensitivity is normally not the sum of local sensitivities). Am I right? If so, RUR does not quantify the relative contribution of the different components to the total uncertainty but it is somehow related to it (it would be interesting to understand how it is).

*AC: We added more information about the RUR (page 9, lines 29-31).*

Page 10, line 13 (and legend of Fig. 4): how can the flood inundation maps be "probabilistic" if the probability of the PMP is unknown? I guess you mean the maps represent estimates of the conditional probability of inundation, conditional to the PMF to have occurred.

*AC: We defined the meaning of this map more precisely in the paper (page 11, lines 12-14).*

Page 13, line 3: do you really think that the total uncertainty range has been quantified in the study? It is to me the range of sensitivity to the different assumptions made.

*AC: We changed this sentence accordingly (page 14, line 14).*

Page 15, line 17: I would say that, given the range of assumptions made here, the

sensitivity of flood losses to the variability of spatial distribution of rainfall is larger than for other factors.

*AC: We changed this sentence accordingly (page 17, lines 16-17.*

Page 16, line 2: could the superimposition of flood waves be quantified for your scenarios?
*AC: We added a subplot in figure 3 to show the contributions of the tributaries to the peak discharge of the Aare River at Bern and added two sentences (page 11, lines 11-13).*

Reviewer #2:

I think that the manuscript presents an important piece of research work which helps to enhance the understanding of several effects on flood loss estimation using the model chain approach which consisting of five components starting with a precipitation module based on PMF approach and ending with a damage module.

The paper is in general well-structured and presented in a quite clear manner. Readers, at least like me, feel comfortable to follow and understand the story of the study reading the excellent introduction chapter. However, going through the remaining chapters I have several points which need to be addressed and clarified by the authors:

- P6, L4-11: Pls explain what "nested" means and why we need this in this context?
*AC: We described the nesting approach in the manuscript* more precisely (page 6, lines 21-32).

P6, L6-8: how about the downstream boundary condition? Additionally, what shown in Figure 2 is quite confusing: why do we need the 2D model when we have already 1D model covering the whole flood plain? How to define coupling points (denoted as black triangles in the figure).

*AC: We described the approach to define the coupling points in the manuscript* (page 6, lines 14-21).

- Figure 1 is not clear and should be improved: (1) at the curved arrow line "boundary conditions", where does the bunch of hydrographs come from? (2)pls expand the figure for the flood loss component adding info on the use of *fdm* and *wse* maps and the five vulnerability functions (V1…V5) because they are used intensively in subsequent plots and discussions.

*AC: We adapted Figure 1 accordingly.*

- Have the exposed buildings/residents corresponding to 1D fdm, 1D wse (Figure 5 and Figure 6) been used for any loss calculation? If not, Pls remove them from the two figures.

Additionally, can the authors explain that the distribution of exposed buildings seems to be bimodal in the cases of 2D but unimodal in the cases of 1D?

*AC: We added two subplots to figure 6 for the losses computed by the 1D model. Furthermore, we will explained the form of the distributions in the exposure (page 13, lines 7-11).*

- P12 L6: "At minimum 2423 … maximum 4667 buildings….". Are these numbers correct? I see at least in the 2D fdm case, the maximum value exceeds 5300 buildings.

*AC: We inserted the correct number in the text (page 13, line 6).*

- I may miss some import points when reading the manuscript! The exposed buildings are higher in 2D fdm compared to those of 2D wse (Figure 5), could the authors help explain why their corresponding losses the opposite (Figure 7 and Figure 8). To this end, I think I would be clearer to explain 'fdm' and 'wse' in a better way (meaning P7L19 – P8L8 is not clear enough!).

*AC: We added the explanations needed for interpreting these results (page 13, lines 21-26).*

- P1L15 and elsewhere: "other uncertainties". There are several kind of uncertainties of which some can be quantified the other cannot. Therefore, the use of "other uncertainties" in this sentence is not recommended even though I understand what the authors mean!

*AC: We changed this sentence.*

- Figure 3: To see clearer spatio-temporal pattern, it would be better to add hydrographs at the outlet of some sub-catchments.

*AC: We added an additional subplot in figure 3.*

- P6 L4: delete "from the wse"

*AC: We deleted this.*